

# NS
## Network Stream Tool

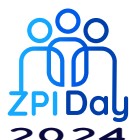

**Autorzy**: Benita Bartczak◉ · Michał Korzeń◉

**Opiekun:** Grzegorz Popek

**Streszczenie**

Celem projektu jest opracowanie oraz implemetacja środowiska eksperymentalnego służącego do szybkiego prototypowania algorytmów strumieniowych. W ciągu dwóch miesięcy udało się stworzyć program, który spełnia najważniejsze funkcjonalności. Użytkownik może wprowadzić własny algorytm strumieniowy, który zostanie uruchomiony na wybranym zbiorze danych. Program pozwala użytkownikowi na skupieniu się na opracowywaniu algorytmów, automatyzując generowanie wykresów oraz niektórych metryk. Wynik prac może być wykorzystany w pracach badawczych w celu uefektywnienia przepływu pracy.

## 1 WSTĘP

Istniejące frameworki przeznaczone do strumieniowego przetwarzania danych wymagają dużego nakładu czasu na naukę technologii oraz konfigurację potoków danych. Są one nastawione na użycie komercyjne. Z kolei mniejszym bibliotekom brakuje automatyzacji rutynowych zadań jak np. obliczanie metryk.

Celem projektu jest stworzenie prostego i szybkiego systemu do przetwarzania danych sieci. Środowisko eksperymentalne ma służyć w celu zwiększenia efektywności pracy badawczej polegającej na testowaniu i prototypowaniu algorytmów wykorzystujących podejście strumieniowe. Będzie zapewniać symulację ciągle napływających nowych danych bez konieczności stosowania bardziej zaawansowanych technologii. Zapewni również możliwość wprowadzenia własnych algorytmów, porównania wyników i metryk, generowania wykresów, dobrania własnego zbioru danych oraz obróbki danych.

## 2 PRACE ZWIĄZANE Z TEMATEM

Wsadowe algorytmy do przetwarzania danych sieciowych (grafowych) nie są dostosowane do wymagań, które stawiają choćby sieci społeczne [1]. Ich głównym aspektem jest stale zmieniająca się struktura – ciągle dołączane są nowe interakcje oraz przerywane są stare połączenia. Dlatego prowadzone są badania nad strumieniowymi algorytmami [2], które określałyby właściwości dynamicznych sieci. Do prowadzenia takich badań potrzebne jest stworzenie odpowiedniego środowiska eksperymentalnego.

Główne zagadnienie projektu dotyczy pracy z takimi danymi w sposób efektywny dla użytkownika, skupiony na szybkim prototypowaniu algorytmów strumieniowych. Zatem nasuwa się pytanie: czy istnieją rozwiązania, które spełniałyby taką potrzebę użytkownika?

Pierwsze szybkie wyszukanie frameworków do pracy ze strumieniami wskazuje na projekt `Kafka` [3]. Projekt ten opiera się o budowanie komunikacji typu wydawca-subskrybent. Skupiony jest wokół tworzenia złożonych systemów obejmujących wiele urządzeń, serwerów czy też rejonów chmurowych. Omawiane zagadnienie dotyczy jednak analizy danych w czasie rzeczywistym a nie ich zbierania.

Zatem bardziej odpowiednie byłoby spojrzenie na bibliotekę `Kafka Streams API` [4], należącą do tego samego zespołu projektów. Skupia się ona na budowaniu aplikacji użytkownika opartej na danych zbieranych strumieniowo. Jednakże wymaga to połączenia z brokerem lub serwerem testowym, które udostępniły by dane programowi. Jest to podejście nieporęczne, jeżeli użytkownik końcowy po prostu chce otworzyć strumień do jednego pliku np. `CSV`. Jest to podstawowa funkcjonalność oferowana przez każdy pełnoprawny język programowania.

Wyżej wymienione technologie są bardzo rozbudowane, także szczególnie nastawione na użycie komercyjne. Grupą docelową projektu `Network Stream Tool` są użytkownicy wykonujący pracę naukową opartą na szybkim prototypowaniu i porównywaniu algorytmów. Tego typu oprogramowanie jest przesadnie duże w stosunku do potrzeb projektu.

Bliższe takiemu kontekstowi naukowemu byłby biblioteki programistyczne takie jak `streamz` oraz `tributary` dla `Pythona`. Oferują one pewną warstwę abstrakcji dla strumieni. Ich zaletą w porównaniu

do wcześniej wymienionych projektów komercyjnych jest to, że nie wymagają np. uruchamiania osobnego serwera, aby otworzyć strumień. Jednakże są to jedynie biblioteki, a nie pełnoprawne środowiska eksperymentalne, które należałoby oddzielnie zaimplementować. Ponadto, rozważając jedynie otwieranie strumienia do zadanego pliku, narzędzia jakie oferują te biblioteki nie wprowadzają znaczących korzyści dla projektu, dodając jedynie niepotrzebną warstwę abstrakcji.

Można także spojrzeć na narzędzia prezentowane głównie jako wynik prac naukowych. Są to zazwyczaj o wiele bardziej wyspacjalizowane frameworki przeznaczone do struktur grafowych. Jednym z nich jest `GraphOne` [5], który oferuje np. przeprowadzanie współbieżnej analityki strumieniowej czy gotowe do testowania zaimplemtowane algorytmy grafowe. Tego typu programy są jednak zdecydowanie poza zakresem prowadzonej pracy, gdzie nacisk kładziony jest na prostotę ze względu na ograniczenia czasowe oraz wielkości zespołu.

Postanowiono projekt oprzeć przede wszystkim na funkcjonalnościach oferowanych domyślnie przez wyznaczony język `Python`. Wybrano go, gdyż posiada duże wsparcie dla analizy danych oraz zespół ma doświadczenie w używaniu go w takim kontekście.

Wszystko, co dotychczas zostało omówione, dotyczy jedynie części warstwy logiki programu. Inna kwestią do rozważenia był sposób zaimplmenetowania interfejsu użytkownika. Jedną z najbardziej znanych bibliotek dla tworzenia interfejsu użytkownika dla języka `Python` to biblioteka `PySide`. PySide jest jednak narzędziem o przeznaczeniu ogólnym - można budować w oparciu o niego róznego rodzaju aplikacje desktopowe. Projekt dotyczy zagadnień z dziedziny analizy danych, zatem warto byłoby wykorzystać gotowe rozwiązania, które są do niej przeznaczone. W szczególności biorąc pod uwagę, że zespół pracuje nadzwyczajnie w składzie 2-osobowym, wskazane jest bazowanie na wyspecjalizowanych rozwiązaniach. W takim ujęciu postanowiono zaimplemetować interfejs z pomocą frameworku `Shiny`, który został stworzony dla użytkowników pracującymi z danymi.

Jednym z nietechnicznych problemów, jakie pojawiły się a początku projektu, było zmniejeszenie grupy do 2 osób. Stworzyło to ryzyko, że nie uda się zrealizować całości planowanych na tamten moment prac. Jako rozwiązanie przede wszystkim postanowiono ograniczyć początkowy zakres projektu - z funkcjonalności usunięto ekstrakcję danych z internetowych repozytoriów danych sieciowych. W wyniku dyskusji ustalono także priorytety funkcjonalności, aby wersja końcowa posiadała najważniejsze podstawowe aspekty aplikacji.

## 3  WYNIKI

W ramach prac prowadzonych w podejściu zwinnym w ciągu dwóch miesięcy udało się zaimplementować najwazniejsze funkcjonalności w postaci:

- Uruchomienia predefiniowanych algorytmów;

- Podania przez użytkownika własnych algorytmów strumieniowych i wsadowych;

- Wczytania zbioru danych i uruchomienie podanego algorytmu na tych danych;

- Zapisywania wyników eksperymentu do pliku (`LaTeX`, `Markdown`);

- Procedur wstępnej obróbki danych;

- Generowania wykresów oraz wartości metryk obrazujących jakość działania algorytmu;

Dodatkowo wdrożono projekt, wykorzystując `GitHub Actions` jako narzędzie ciągłej integracji. Automatyzuje ono sprawdzanie jakości kodu oraz budowanie paczki z plikiem wykonywalnym jako gotowe wydanie aplikacji.

W praktyce wytworzony program może być używany przez dowolną osobę, która chciałaby szybko sprawdzić efektywność algorytmu strumieniowego.

## 4  WNIOSKI

Obecna wersja projektu jest w stanie zwiększyć efektywność pracy związanej z prototypowaniem algorytmów strumieniowych. Oferuje automatyzację pewnych żmudnych zadań, które musi wykonać pracownik badawczy zajmujący się tematem algorytmów strumieniowych. Takimi rutynami są np. otwarcie strumienia do pliku (`CSV`, `MTX`, `TXT`), generowanie wykresów (czasu działania algorytmu, zużycia pamięci) oraz metryk jakości rozwiązania (takich jak podobieństwo Jaccarda). Najważniejszymi sukcesami projektu są możliwość podania własnego algorytmu przez użytkownika do narzędzia oraz uruchomienie go na zadanym zbiorze danych.

## 4.1 Kierunki rozwoju

Projekt w obecnej wersji mógłby zostać rozszerzony o funkcjonalność polegającą na możliwość podania przez użytkownika własnych metryk, które go interesują, i które chciałby, żeby były obliczane i wyświetlane przez aplikację. Dodatkowo, za względu na to, że program jest uruchamiany lokalnie na maszynie użytkownika, warto byłoby dodać opcję uruchamiania wielu iteracji tego samego eksperymentu. W ten sposób wyniki byłyby bardziej wiarygodne i mniej zależne od problemów np. związanych ze systemem operacyjnym. Ponadto wskazane byłoby dodanie trybu debugowania dla algrorytmów dostarczonych przez użytkownika.

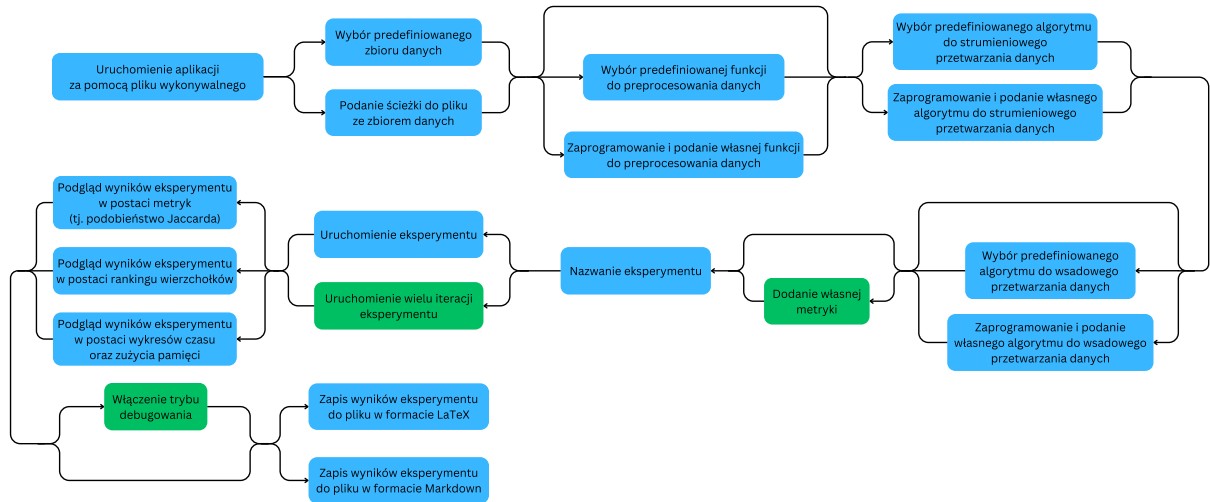

Rysunek 1: Workflow aplikacji (zielone bloczki wskazują na dalsze kierunki rozwoju)

# LITERATURA

[1] Andrew McGregor. Graph stream algorithms: a survey. *SIGMOD Rec.*, 43(1), 2014.

[2] Maciej Besta, Marc Fischer, Vasiliki Kalavri, Michael Kapralov, and Torsten Hoefler. Practice of streaming processing of dynamic graphs: Concepts, models, and systems, 2021.

[3] Apache Software Foundation. Kafka introduction. `https://kafka.apache.org/intro`. Wersja z dnia 20 października 2024r.

[4] Apache Software Foundation. Kafka streams documentation. `https://kafka.apache.org/38/documentation/streams/tutorial`. Wersja z dnia 20 października 2024r.

[5] Pradeep Kumar and H. Howie Huang. Graphone: A data store for real-time analytics on evolving graphs. *ACM Trans. Storage*, 15(4), January 2020.
