# OpenReview forum: "Network Stream Tool"
_pwr.edu.pl/Wrocław_University_of_Science_and_Technology/2024/ZPI_Day — Wrocław University of Science and Technology 2024 ZPI Day Submission_

### Official Review · Reviewer_NcJB · 2024-12-06
**Network Stream Tool Review**

**Confidence:** 5
**Significance Of Results:** 4
**Overall Quality:** 4

**Compliance With Template:**

5: Very High Quality – The article contains all the required sections, which are written in a very detailed, clear, and error-free manner. The structure is professional and meets expectations, and the content adheres to the highest substantive and formal standards.

**Description Of Results:**

4: High Quality – The results are described in detail and supported by usage examples or evaluations. The description is reliable but may lack full depth of analysis.

**Feedback On Consistency:**

Nakreślenie tematyki projektu i jego celu jest poprawne i odnosi się do rzeczywistego problemu w dziedzinie badań nad dynamicznymi strukturami grafowymi.
Odniesienia do istniejących rozwiązań (i nowinek w tym zakresie) nie były liczne, lecz należy pamiętać, że nie jest to projekt ściśle badawczy.
Workflow aplikacji jest przedstawiony dość późno w tekście. Dopiero wtedy widzimy strukturę obsługiwanego procesu, wcześniej jego elementy są opisywane dość pobieżnie. Moim zdaniem wokół tego rysunku powinna być kreowana główna narracja dotycząca sposobu realizacji poszczególnych etapów/elementów/funkcjonalności w projekcie.

---
Uwagi edycyjne: tekst zawiera nieliczne literówki (np. zaimplmenetowania, zmniejeszenie)

---
Uwagi dotyczące stylu komunikacji: Rozumiem, że grupa miała zmniejszoną liczność, ale wystarczyło powiedzieć to raz, a było to jeszcze dość nachalnie powtarzane w ostatnim paragrafie sekcji 2.

**Potential For Development:**

Opracowanie ma potencjał rozwojowy. Należy jednak zwrócić uwagę na to, że na ten moment autorzy nie zwrócili jeszcze uwagi na zagadnienia optymalizacyjne wdrażanych algorytmów bazowych i miar. Może to mieć istotny wpływ na wydajność w przypadku prób wykorzystania rozwiązania dla większych struktur grafowych.

**Project Nature Evaluation:**

W ramach projektu opracowywane jest analityczne środowisko programistyczne, organizujące i częściowo automatyzujące pracę analityka danych.

**Technical Language Precision:**

4: High Quality – The language is appropriate for a technical report. Terminology is used correctly, and statements are precise, with only minor shortcomings that do not affect the overall clarity.

---

### Official Review · Reviewer_EhTj · 2024-12-06
**Network Stream Tool**

**Confidence:** 2
**Significance Of Results:** 3
**Overall Quality:** 3

**Compliance With Template:**

3: Average Quality – The article includes most of the required sections, but some may be incomplete, written in a general or unclear manner. The content is correct but requires further refinement.

**Description Of Results:**

2: Low Quality – The results are described very superficially and in a general manner. Essential details, usage examples, or evaluations are missing.

**Feedback On Consistency:**

Struktura (kolejność sekcji) dokumentu jest poprawna, jednak przejścia pomiędzy kolejnymi sekcjami nie są jasne i nie tworzą spójnej całości. Opis wyników jest niekompletny - zawiera jedynie wymienione funkcjonalności, które zostały zaimplementowane. Brakuje prezentacji wyników i efektów działania zaimplementowanego narzędzia.

**Potential For Development:**

W artykule zostały przedstawione potencjalne możliwości rozwoju.

**Project Nature Evaluation:**

Projekt wykazuje cechy pracy inżynierskiej.

**Technical Language Precision:**

4: High Quality – The language is appropriate for a technical report. Terminology is used correctly, and statements are precise, with only minor shortcomings that do not affect the overall clarity.

---

### Official Review · Reviewer_Nxon · 2024-12-06
**Network Stream Tool**

**Confidence:** 4
**Significance Of Results:** 3
**Overall Quality:** 3

**Compliance With Template:**

3: Average Quality – The article includes most of the required sections, but some may be incomplete, written in a general or unclear manner. The content is correct but requires further refinement.

**Description Of Results:**

3: Average Quality – The results are described with moderate detail. Some examples or evaluation elements are present but insufficiently developed or incomplete.

**Feedback On Consistency:**

Jak na niecałe trzy strony tekstu, to zauważyć można jednak zbyt dużo błędów edycyjnych:
  udostępniły by dane
  takżę
  byłby biblioteki
  z pomocą frameworku --> za pomocą frameworku
  pojawiły się a początku projektu
  do 2 osób --> do dwóch osób
  w postaci: • Uruchomienia ... jakość działania algorytmu; --> w postaci: • uruchomienia ... jakość działania algorytmu.
  związanych ze systemem
oraz wiele błędów interpunkcyjnych.

**Potential For Development:**

Sądząc na podstawie tego dość krótkiego opisu. krótkiego gdyż dopuszczalne były cztery strony a wykorzystano niecałe trzy strony, zaimplementowana aplikacja wydaje się być dobrą podstawą do dalszych prac w tym zakresie.

**Project Nature Evaluation:**

Zespół pracował w składzie 2-osobowym, trudno zatem mówić w tym przypadku o przedsięwzięciu zespołowym.
Na takie przedsięwzięcie inżynierskie nie powinno być zgody, gdyż jego realizacja nie pozwoliła na osiągnięcie efektu umiejętności pracy w zespole
Bardzo ciekawym elementem, zrealizowanym według Autorów, w ramach zaproponowanego i zaimplementowanego narzędzia jest możliwość podania przez użytkownika własnych algorytmów strumieniowych i wsadowych. Szkoda, że w opracowaniu nie wskazano jak to jest możliwe i jak to zostało wykonane.

**Technical Language Precision:**

4: High Quality – The language is appropriate for a technical report. Terminology is used correctly, and statements are precise, with only minor shortcomings that do not affect the overall clarity.

---

### Decision · Program_Chairs · 2024-12-10

Accept (Poster)